# Panoptic Segmentation on Panoramic Radiographs: Deep Learning-Based Segmentation of Various Structures Including Maxillary Sinus and Mandibular Canal

**DOI:** 10.3390/jcm10122577

**Published:** 2021-06-11

**Authors:** Jun-Young Cha, Hyung-In Yoon, In-Sung Yeo, Kyung-Hoe Huh, Jung-Suk Han

**Affiliations:** 1Department of Prosthodontics, School of Dentistry and Dental Research Institute, Seoul National University, Daehak-ro 101, Jongro-gu, Seoul 03080, Korea; starriet@snu.ac.kr (J.-Y.C.); drhiy226@snu.ac.kr (H.-I.Y.); pros53@snu.ac.kr (I.-S.Y.); 2Department of Oral and Maxillofacial Radiology, School of Dentistry and Dental Research Institute, Seoul National University, Daehak-ro 101, Jongro-gu, Seoul 03080, Korea

**Keywords:** panoptic segmentation, semantic segmentation, instance segmentation, object detection, deep learning, machine learning, artificial intelligence, dental panoramic radiograph

## Abstract

Panoramic radiographs, also known as orthopantomograms, are routinely used in most dental clinics. However, it has been difficult to develop an automated method that detects the various structures present in these radiographs. One of the main reasons for this is that structures of various sizes and shapes are collectively shown in the image. In order to solve this problem, the recently proposed concept of panoptic segmentation, which integrates instance segmentation and semantic segmentation, was applied to panoramic radiographs. A state-of-the-art deep neural network model designed for panoptic segmentation was trained to segment the maxillary sinus, maxilla, mandible, mandibular canal, normal teeth, treated teeth, and dental implants on panoramic radiographs. Unlike conventional semantic segmentation, each object in the tooth and implant classes was individually classified. For evaluation, the panoptic quality, segmentation quality, recognition quality, intersection over union (IoU), and instance-level IoU were calculated. The evaluation and visualization results showed that the deep learning-based artificial intelligence model can perform panoptic segmentation of images, including those of the maxillary sinus and mandibular canal, on panoramic radiographs. This automatic machine learning method might assist dental practitioners to set up treatment plans and diagnose oral and maxillofacial diseases.

## 1. Introduction

Deep convolutional neural networks, one of the major examples of deep learning, aid in image classification and in predicting the location of the region of interest (RoI) in images. Several studies have sought to apply these deep neural networks on medical images; the application of semantic segmentation has been attempted in some of these studies [1,2,3] where each pixel in the image is classified. Alternatively, other studies have tried methods such as object detection [4,5] or instance segmentation [6]. Studies on applying convolutional neural networks (CNNs) to radiographs are increasing rapidly in the field of dentistry, and many of them deal with panoramic radiographs [7]. The application of semantic segmentation [8], object detection [9,10,11], and instance segmentation [12] on dental panoramic radiographs (orthopantomogram) has been reported.

Panoramic radiographs are commonly used during conventional dental treatment. Therefore, it would be useful to automatically detect and segment different structures of multiple categories on the dental panoramic radiograph for the diagnosis and treatment planning of various diseases. However, to the best of our knowledge, the use of a machine learning method for this kind of task, including deep neural networks, has not been studied in the dental field so far.

It is difficult to apply an artificial intelligence method to panoramic radiographs for detecting and segmenting different kinds of structures because structures of various sizes and shapes are overlapped together in the panoramic radiograph. In some cases, a ghost or double image interferes with the identification of a specific structure.

Moreover, the boundaries of some important structures, such as the mandibular canal or maxillary sinus, are difficult to distinguish in the panoramic radiograph. For a successful dental implant surgery, it is important to identify the boundaries of the maxillary sinus and mandibular canal to determine the surgical method and the type of implant to be used. Nonetheless, no previous studies have attempted to automatically segment these structures on a panoramic radiograph.

Therefore, this study aimed to automatically segment various types of structures, including the maxillary sinus and mandibular canal, in an orthopantomogram. The following classes were detected and segmented: maxillary sinus, maxilla, mandibular canal, mandible, normal tooth, treated tooth, and dental implants.

A new concept called “panoptic segmentation,” which was recently proposed to integrate different tasks in computer vision, was applied in the current study [13]. This task combines semantic segmentation and instance segmentation and requires predicting not only the semantic label but also instance id for each pixel. Regions that are not countable, such as grass or road, must be segmented by semantic label classification on each pixel, as in the semantic segmentation task, whereas countable objects, such as a person or car, must be segmented by both semantic label and instance id classification on each pixel, which produces results similar to those of the instance segmentation task.

Panoptic segmentation is one of the most challenging tasks in the field of computer vision; hence, a recently developed and verified deep convolutional neural network model designed for panoptic segmentation was adopted [14]. This deep learning-based automatic method might assist dental practitioners in developing appropriate treatment plans and diagnosing various oral and maxillofacial diseases.

## 2. Materials and Methods

### 2.1. Ethics Statement

This study was conducted in accordance with the deliberation results by the Institutional Review Board of Seoul National University Dental Hospital, Dental Life Science Research Institute (IRB No. ERI20024, Notification of deliberation exemption on August 2020). Ethical review, approval, and patient consent were waived for this study owing to the following reasons: this study is not a human subjects research project specified in the Bioethics and Safety Act, it is practically impossible to obtain the consent of the research subjects in this study, and the risk to the subjects is extremely low because of the retrospective nature of the study.

### 2.2. Datasets

Ninety dental panoramic radiographs were obtained from the Picture Archiving and Communication System in the Seoul National University Dental Hospital. In some cases, it was not possible to annotate all the regions accurately because the radiographs were blurred, or the structures overlapped excessively. Those radiographs were not included in the study since it is difficult to create the correct annotations. For example, the boundary of the medial wall of the maxillary sinus might be unclear, the border of the mandibular canal might not be visible, the patient might not be properly positioned inside the focal trough, or some radiopaque materials might hinder the discrimination of the jaw structures. In addition, radiographs of patients who had undergone unusual treatments were excluded since those cases are not sufficient and it is difficult to make a clear annotation for those radiographs. For example, the panoramic radiographs including mandibular reconstruction, Caldwell-Luc operation, or LeFort II/III surgeries were excluded. When the Caldwell-Luc operation was performed on the maxillary sinus, the panoramic image was excluded because the cortical bone line of the maxillary sinus wall is not properly displayed and changed into an abnormal pattern in the panoramic image due to atrophy of the maxillary sinus. Moreover, when LeFort II or III surgery was performed on the maxilla, the panoramic radiographs of the patients with maxillary reconstruction were excluded because it affects the maxillary sinus and makes annotation difficult. In the case of chronic sinusitis, thickening and blurring of the maxillary sinus wall occur, and the panoramic radiograph was excluded if it was not possible for a dentist to annotate the outline of the maxillary sinus.

All of the panoramic radiographs were acquired by using the same digital panoramic machine (OP100, Instrumentarium Corp., Tuusula, Finland) at 66~73 kVp, 6.4~12 mA, and an exposure time of 16.8~17.6 s. X-ray imaging conditions changed depending on the patient’s body size and degree of obesity. Patients were positioned in the panoramic machine according to the manufacturer’s recommendations, and several radiologists took the images. All images were obtained using PSP image plates (12 × 10 inch) and read by an FCR system (Fuji Computed Radiography 5000R, Fuji Photo Film Co. Ltd., Düsseldorf, Germany).

Finally, a total of 51 panoramic radiographs were randomly separated into 3 groups: training (*n* = 30), validation (*n* = 11), and test (*n* = 10) datasets. The age of the patients ranged from 32 to 79 years with a mean age of 60.6 years. Among the patients, 35 were male and 16 were female. The ground truths were annotated by a dental practitioner (J.-Y.C.), and an oral and maxillofacial radiologist (K.-H.H.) reviewed, corrected, and confirmed the annotations. Visualized examples of the annotations are shown in Figure 1.

For the panoptic segmentation task, the classes that are subjected to semantic segmentation are referred to as “stuff”, whereas those subjected to instance segmentation are referred to as “thing” [13]. The eight classes (five stuff and three things) included in the current study were as follows: maxilla, maxillary sinus, mandible, mandibular canal, normal tooth, treated tooth, dental implant, and unlabeled. Among these, normal tooth, treated tooth, and dental implant were assigned as things; thus, each object in these classes was segmented individually.

Some classes were categorized to help understand the results easily: the maxilla and mandible were categorized as “bone”, whereas the normal tooth, treated tooth, and dental implant were categorized as “tooth”. The maxillary sinus and mandibular canal were not categorized because they did not share a similar morphology with the other classes.

### 2.3. Neural Network Model Architecture

Panoptic segmentation is challenging and has not been studied on panoramic radiographs; therefore, it is important to use a high-performance artificial neural network model. The results of the Cityscapes dataset benchmark [15] were investigated to select a proper state-of-the-art deep neural network. On the basis of the evaluation results of various models, one of the most high-performance models, the Panoptic DeepLab [14], was selected. It consists of a semantic segmentation branch and an instance segmentation branch, both of which share the same encoder as the backbone. The instance segmentation branch predicts the center of the mass for each instance and the offset vector, which starts from each foreground pixel and points to the corresponding center of mass.

On the basis of the center and the offset vector, the instance ID of each foreground pixel can be decided. Each pixel is relocated by its offset vector, and the distance between the predicted instance center and the relocated pixel is calculated. Then, the index of the closest instance center is allocated to the pixel for its instance ID, which yields the result of the instance segmentation branch. By merging the prediction result of the semantic segmentation branch with that of the instance segmentation branch, the final panoptic segmentation result can be obtained. The overview of the model is shown in Figure 2.

### 2.4. Evaluation Methods

Panoptic segmentation encompasses both semantic and instance segmentation; thus, the inference results of the model in this study can be evaluated from several different perspectives. Taking all these perspectives into account, the metrics designed for panoptic segmentation and those for semantic and instance segmentation were used in this study.

First, the panoptic quality (PQ), segmentation quality (SQ), and recognition quality (RQ) were obtained, as proposed in a previous study [13]. These metrics consider the semantic and instance perspectives in a comprehensive manner and are widely used for the evaluation of the results from the panoptic segmentation [15,16].

PQ is defined as
(1)$$\mathrm{PQ}=\frac{\sum_{\left(p,g\right)\in \mathrm{TP}}\mathrm{IoU}\left(p,g\right)}{\left|\mathrm{TP}\right|+\frac{1}{2}\left|\mathrm{FP}\right|+\frac{1}{2}\left|\mathrm{FN}\right|}$$
where *p*, *g*, and IoU represent the predicted segment, ground truth segment, and intersection over union (IoU), respectively. TP, FP, and FN represent the true positives, false positives, and false negatives, respectively, at the instance level. In more detail, TP, FP, and FN are matched segment pairs, unmatched predicted segments, and unmatched ground truth segments, respectively, found by segment matching after calculating the IoU. PQ is calculated for each class and can be averaged over classes, as shown in the result section.

In Equation (1), IoU, which is also known as the Jaccard index, is defined as
(2)$$\mathrm{IoU}\left(p,g\right)=\frac{\left|p\cap g\right|}{\left|p\cup g\right|}$$
where *p* and *g* represent the predicted segment and ground truth segment, respectively. Only segment pairs with IoUs greater than 0.5 were considered to be matched pairs.

Moreover, PQ can be expressed as a product of SQ and RQ:(3)PQ=SQ×RQ
where SQ is defined as
(4)$$\mathrm{SQ}=\frac{\sum_{\left(p,g\right)\in \mathrm{TP}}\mathrm{IoU}\left(p,g\right)}{\left|\mathrm{TP}\right|}$$
and RQ as
(5)$$\mathrm{RQ}=\frac{\left|\mathrm{TP}\right|}{\left|\mathrm{TP}\right|+\frac{1}{2}\left|\mathrm{FP}\right|+\frac{1}{2}\left|\mathrm{FN}\right|}$$

In Equation (4), SQ is the averaged IoU of all the matched segment pairs. Furthermore, in Equation (5), RQ is the same as the F1 score, the harmonic mean of precision and recall. Thus, decomposing PQ into these two terms helps in interpreting the PQ.

The IoU and instance-level IoU (iIoU) [17] were calculated to evaluate the model’s inference results from the perspective of the pixel-level semantic segmentation. Unlike the IoU in Equation (1), where each IoU was calculated for each segment pair, the IoU, in this case, was calculated for each class, globally across all the panoramic radiographs in the test dataset:(6)$$\mathrm{IoU}=\frac{\mathrm{TP}}{\mathrm{TP}+\mathrm{FP}+\mathrm{FN}}$$

The essential concept of IoU is the same in Equations (2) and (6), but we used a different notation to emphasize the difference described above. Unlike in Equations (1), (4) and (5), the TP, FP, and FN in Equation (6) represent the number of pixels of true positive, false positive, and false negative, calculated for one class summed over all the panoramic radiographs.

However, the IoU in Equation (6) has some bias toward large objects. To address this, iIoU uses values that are adjusted with the scale of each object:(7)$$i\mathrm{IoU}=\frac{i\mathrm{TP}}{i\mathrm{TP}+\mathrm{FP}+i\mathrm{FN}}$$
where iTP and iFN are, respectively, TP and FN weighted by the ratio of the average instance area of the class to the area of each ground truth instance:(8)$$iTP=\sum_{i}\left[TP_{i}\times \frac{\left(\mathrm{average}\;\mathrm{instance}\;\mathrm{area}\;\mathrm{of}\;\mathrm{the}\;\mathrm{class}\right)}{\left(\mathrm{area}\;\mathrm{of}\;\mathrm{the}\;\mathrm{ground}\;\mathrm{truth}\;\mathrm{instance}\;i\right)}\right]$$
(9)$$iFN=\sum_{i}\left[FN_{i}\times \frac{\left(\mathrm{average}\;\mathrm{instance}\;\mathrm{area}\;\mathrm{of}\;\mathrm{the}\;\mathrm{class}\right)}{\left(\mathrm{area}\;\mathrm{of}\;\mathrm{the}\;\mathrm{ground}\;\mathrm{truth}\;\mathrm{instance}\;i\right)}\right]$$

In Equations (8) and (9), ${\mathrm{TP}}_{i}$ and ${\mathrm{FN}}_{i}$ represent the number of pixels of true positive and false negative, respectively, that belong only to the corresponding ground truth instance *i*.

Since this metric assumes that the model’s output does not include any information about distinguishing among the individual instances, the pixels that correspond to FP do not belong to a specific instance. Thus, FP is not weighted. Moreover, the IoU and iIoU were calculated for each class and each category.

Lastly, to evaluate the model’s inference results from the perspective of the instance segmentation, the average precision (AP) for each “thing” was calculated and averaged across 10 different IoU threshold values ranging from 0.5 to 0.95 in steps of 0.05, as it is a widely used method for avoiding bias toward a specific threshold [18].

### 2.5. Neural Network Training Specifications

For transfer learning, ImageNet [19] pre-trained weights were used. The training data were enriched using randomly generated data augmentation, which included both horizontal flipping and randomized cropping. The hyperparameters of the model, such as base learning rate or the number of total iterations, were chosen after investigating the evaluation results of the validation dataset. Consequently, the base learning rate was set to 0.001, warmed up for 1000 iterations, and gradually decreased as the iteration progressed. The total iteration was 65,000, and the Adam method [20] was used for neural network optimization. The neural network model was trained on a cloud machine (Colaboratory, Google Research, Mountain View, CA, USA) with a 16 GB GPU accelerator (Tesla V100, Nvidia, Santa Clara, CA, USA).

The software codes for preprocessing the data, training and running the model for inference results, and computing the evaluation metrics were mostly programmed with Python (Python 3.7.10, Python Software Foundation, Wilmington, DE, USA). To facilitate the model training and inferencing, an open-source project [21] was modified and used as a library, based on a machine learning library with GPU support (PyTorch 1.8.1, Facebook AI Research, Menlo Park, CA, USA). An annotation tool [22] was used to prepare the datasets, and the results were preprocessed to proper formats in order to be fed into the model. The model used in this study was adopted from a previous study [14]. In addition, evaluation metrics from previous studies [13,17] and the related open-source codes [23,24] with some modifications were used.

## 3. Results

### 3.1. Visualization of the Inference Results

In order to visually examine the inference results, the output values of the model were visualized and superimposed on the original inputs of the panoramic radiographs (Figure 3).

### 3.2. Evaluation with Panoptic Segmentation Metrics

The metrics for panoptic segmentation, PQ, SQ, and RQ, were computed for each class and averaged over three different categories: all classes, things, and stuff. The PQ, SQ, and RQ values averaged over all classes were 74.9, 83.2, and 90.0, respectively. The evaluation results are presented in Table 1.

### 3.3. Evaluation with Semantic Segmentation Metrics

The metrics for pixel-level semantic segmentation, IoU and iIoU, were computed for each class and category. The iIoU was computed only for “things” and the corresponding category. In addition, a confusion matrix is presented to simplify the evaluation results (Figure 4). In the confusion matrix, each row represents the ground truth class, whereas each column represents the predicted class of the model. Each value in a cell represents a ratio of the number of pixels predicted by the model (as a class of the column among the pixels of the ground truth class) to the number of pixels of the ground truth class. A prior was computed for each row in the matrix, which represented a ratio of the number of pixels of the corresponding ground truth class to the total number of pixels. The confusion matrix and the IoU and iIoU for each class and category are shown in Figure 4, Table 2 and Table 3, respectively.

### 3.4. Evaluation with Instance Segmentation Metrics

The metrics for instance segmentation, AP, were computed for each thing. The AP values averaged over all the IoU thresholds for the normal tooth, treated tooth, and dental implant were 0.520, 0.316, and 0.414, respectively (Table 4). Additionally, the AP value at the IoU threshold of 0.5, which is widely used for evaluation in the object detection task, has been presented for ease of reference.

## 4. Discussion

Dental panoramic radiographs are used to detect and diagnose diseases of the oral and maxillofacial area and to create various treatment plans in the dental clinic. Therefore, it is important to accurately read a panoramic radiograph in the field of dentistry.

Several studies on artificial intelligence models targeting dental panoramic radiographs have been published in the past. Some of them aimed to binary-classify the presence of specific diseases [25] or classify the types of diseases using panoramic images [26]; however, the deep neural network models used in these studies could classify images but not locate the disease within the panoramic radiographs. Moreover, most of these models needed to be trained with cropped images, and the cropping should be done manually so that the RoI is located in the center of the cropped image.

On the other hand, some studies attempted to perform object detection, which predicts the location of the disease and classifies it, using panoramic radiographs [9,10,11]. Furthermore, instance segmentation, which not only locates the object or lesion but also segments its outline, was applied using panoramic radiographs [12]. However, most instance segmentation models were not designed to segment extremely wide or long objects, such as the jaw or mandibular canal shown in panoramic radiographs. For example, one of the most widely studied and used convolutional neural networks (CNNs) for instance segmentation, the Mask R-CNN [27], uses anchors of various sizes and aspect ratios to predict the RoI before further regression. Although it is possible to adjust the size, aspect ratio, and angle of the anchor to fit the jaws or mandibular canal in the panoramic radiograph, the model was not designed to detect such unusual objects in the first place. Some studies applied semantic segmentation [8], which classifies each pixel but does not distinguish between each tooth on the panoramic radiograph.

Each of these approaches has its advantages and disadvantages. Panoptic segmentation was recently proposed to combine the different types of tasks [13], and deep learning models that can perform this task are currently being evaluated [14,28]. However, to the best of our knowledge, panoptic segmentation has not been performed in the fields of medicine and dentistry.

In the current study, a state-of-the-art deep learning model capable of panoptic segmentation was applied to dental panoramic radiographs. Remarkable results were obtained, as observed from the evaluation and visualization results. It is difficult to distinguish the various structures and the double as well as ghost images visible on the panoramic radiograph. The outlines of the maxillary sinus and mandibular canal are often difficult to find, even for an experienced dentist. However, it is important to identify the boundaries of these structures, particularly during treatment planning. Therefore, unlike many previous studies that mainly focused only on teeth segmentation [8,12], the present study examined whether the maxillary sinus and mandibular canal can be identified on dental panoramic radiographs using a deep neural network model.

The segmentation of the mandibular canal showed the lowest PQ and SQ among the “stuff” classes, which was consistent with the results of the IoU. However, as shown in Figure 3, most of the original input panoramic radiographs tested were faint, making it difficult to read and identify the mandibular canal. Moreover, as can be seen in the confusion matrix, the prior value of the mandibular canal was the lowest among all classes, which showed that the area covered by the mandibular canals was less than 2% of the entire pixels. Nevertheless, our model correctly detected almost 80% of the ground truth pixels of the mandibular canal. A previous study, which proposed the concept of panoptic segmentation, tried to compare the artificial neural networks to human annotators and showed that human annotators outperformed the machines [13]. Given that artificial intelligence has not surpassed humans in this task so far, the recognition of the mandibular canal in the current study might be considered as a meaningful achievement.

Among the stuff classes, the maxilla showed the second lowest evaluation score after the mandibular canal, which can be seen from the PQ and IoU results. The reason for this is presumed to be the unclear boundary between the maxillary sinus and maxilla; moreover, some structures, such as the hard palate and zygomatic arch, often interfere with the readings. There are cases where the central part of the maxilla is not clear because of the overlap of a ghost image caused by the cervical spine. In addition, the number of pixels in the area covered by the maxilla was the second smallest (<5%) after the number of pixels covered by the mandibular canal, among the stuff classes. Nonetheless, the model detected almost 90% of the ground truth pixels of the maxilla correctly, as can be seen in the confusion matrix. Surprisingly, the results for the maxillary sinus were even better (96% of the total ground truth pixels were accurately detected), considering the fact that the medial wall and the floor of the maxillary sinus are difficult to identify and interfere with other structures, such as the innominate line and nasal floor, in most cases. It is very important to determine the location and shape of the maxillary sinus and mandibular canal during dental implant surgery; therefore, these results suggest that artificial intelligence will be of great help in dental clinics in the future.

In the case of the thing classes, the treated tooth class showed the second lowest IoU after the mandibular canal. This is presumed to be due to the disadvantage in training because the number of treated teeth was relatively smaller than the number of normal teeth. A similar trend was observed with the iIoU and AP values. Moreover, the treated tooth showed the lowest PQ among all classes because of the low RQ. One of the main reasons for this is that many of the treated teeth and dental implants are bridges; thus, it might have been difficult for the model to distinguish between an individual tooth and an implant. In the case of a bridge without a pontic, or a single crown adjacent to a bridge or another single crown, it is sometimes difficult to ascertain whether they are just adjacent or connected to each other.

In addition, in the case of a bridge that connects a natural tooth to a dental implant, the evaluation score is inevitably lowered because the boundary between the treated tooth class and the implant class cannot be distinguished. The presence of a pontic in a bridge that connects the same type of abutment teeth, i.e., only natural teeth or only dental implants, ensures that the abutments are one connected instance. However, when the types of abutments are different, i.e., when a natural tooth is connected to an implant, it is not possible to identify the boundary between the treated tooth class and the dental implant class, despite the presence of a pontic.

Even if a very small segment is detected incorrectly, it can negatively affect the RQ value to the same extent as when a very large segment is incorrectly detected. This is because the segment area itself does not affect the RQ. Unlike normal teeth and dental implants, the treated tooth class covers various treatments, such as fixed dental prostheses, different types of restorations, and root canal treatment. Owing to this characteristic of the treated tooth class, there are several cases where the segments predicted by the model are split and fragmented, thereby reducing the RQ.

These issues seem to have a similar effect on the AP, as can be seen from the low AP values in the treated tooth class. However, these problems do not decrease the IoU, which is calculated regardless of the distinguishing of the individual instances.

The IoU in Equation (6) is a widely used metric in semantic segmentation; however, it does not reflect whether each object is detected, and has bias toward large objects. To alleviate this shortcoming, the iIoU, which normalizes the IoU by the area of each instance, was used in the current study. Nevertheless, the ranking of the thing classes according to the evaluation score was the same for both IoU and iIoU, in this study. This might be attributed to the small differences in the areas between the objects belonging to the thing classes on the panoramic radiographs. If there was a large area difference between individual instances, the iIoU value could have been much different from the IoU value.

The thing classes showed lower PQ values than did the stuff classes, mainly owing to the difference in RQ; in addition, the treated tooth class played a big role, as described earlier. Some degree of loss of RQ was observed even in the normal tooth class, where there was no need to distinguish between the bridge and the adjacent single crowns. One of the reasons for this is probably the nature of the dental panoramic radiograph, wherein multiple teeth of similar shape and size are located close to each other. In many computer vision tasks, if a large number of objects of a particular class are clustered in one place, they are separated into another class because it is difficult to distinguish between each object. Taking this into account, it is quite natural to have a low RQ value for the thing classes, because crowded teeth are frequently observed in dental panoramic radiographs.

Unlike the instance segmentation task, where each segment can overlap other segments, each pixel has only one value in the panoptic segmentation class. This is not a major problem in typical images because if a specific object covers the object behind it, the covered part of the object is not visible. However, in radiographs, even if the object in the front covers the object behind, the object behind can be visible, so a situation arises in which a specific pixel must have multiple values. This situation occurs frequently in normal tooth classes because the crowns are very often overlapped, and the hidden parts are visible. Thus, annotating the ground truth for the normal tooth class requires some compromise. When two crowns overlap each other, the midpoint of the overlapped portion is assumed as the boundary between the two crowns because a pixel can have only one instance ID. Alternatively, it is possible to treat all the teeth as one instance if the crowns overlap. However, this method was not used because, in some cases, it was not clear as to whether the crowns were overlapped. The evaluation index results differ greatly depending on whether the teeth are viewed as one instance or separate instances. This problem could adversely affect not only the RQ but also the SQ of the normal tooth class because SQ can be interpreted as the averaged IoU overall matched segment pairs.

The evaluation metrics for semantic segmentation in our study, such as global IoU and iIoU, are calculated for all pixels in the ground truths and the model’s inferences of the corresponding class in the entire test dataset. Thus, these metrics do not result in multiple values for a specific class. Since the result of the evaluation has only one value, for one class, it is difficult to compare the values from the neural network and clinicians in a statistical manner, as there is no average or standard deviation. Other evaluation metrics, PQ as well as SQ and RQ, are also calculated for each class, not for each instance. Therefore, the same logic applies. In addition, AP is not applicable for humans since plotting precision-recall graphs, which is used for obtaining AP values, requires confidence scores from a neural network model. Therefore, the metrics used in this study are not very good for comparing the model and dental clinicians. As machine learning and deep learning techniques are developing day by day, future studies need to develop and use evaluation methods that are easy to compare models with humans.

It is worth noting that decent results were obtained in the current study, despite using a significantly smaller number of datasets than those used for general machine learning and deep learning training. This might be attributed to the transfer learning, data augmentation, and standardized imaging methods used for dental panoramic radiographs. Unlike general computer vision tasks, in the case of panoramic radiographs, the radiograph is taken in a consistent manner with the patient positioned at a certain location and angle. Moreover, the structures in the image are arranged in a specific pattern. Therefore, considering the number of radiographs used in this study, we believe that better results can be obtained if a larger amount of data is used.

In this study, the panoramic radiographs were taken with a specific machine in one hospital. The brand and company of the machine used to take orthopantomograms greatly affect the quality and characteristics of the radiographs. It has been shown in a previous study that the performance of the model can be improved when panoramic radiographs from different hospitals are mixed (cross-center training) [29]. Therefore, it is possible to develop a more generalized deep neural network by using radiographs taken by various types of panoramic radiograph machines. Further studies are needed to improve generalization and avoid overfitting of neural networks. 

Moreover, several pieces of radiographic equipment that take better quality panoramic radiographs have been developed in recent years, and many of them are being introduced in dental clinics. Given that the panoramic radiographs used in this study have inferior quality compared to those taken with recently developed machines, the inference results will be further improved if the model is trained and tested with higher quality images. Thus, with the development of radiographic equipment, artificial intelligence might be of assistance for the reading of panoramic radiographs in the future.

Although the machine learning method referred to in this study can segment important structures in panoramic radiographs, there are still many structures in the panoramic radiograph that have not been considered in this study. Applying the abovementioned improvements, the method used in this study will be a foundation for future studies on detecting more diverse structures in dental panoramic radiographs.

## 5. Conclusions

The deep neural network model presented in this study, which is designed for panoptic segmentation, could detect and segment various structures in dental panoramic radiographs. The model was able to segment the maxillary sinus and mandibular canal, which are often difficult to distinguish on a radiograph. This automatic machine learning method may assist dental practitioners while setting up treatment plans and diagnosing oral and maxillofacial diseases.

## Figures and Tables

**Figure 1 jcm-10-02577-f001:**
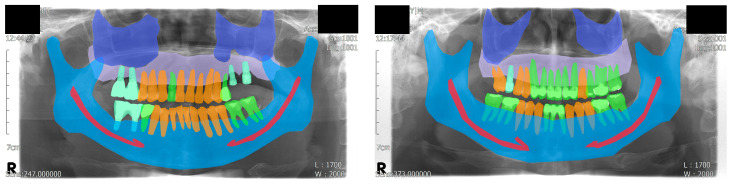
Visualized examples of the annotation results. A total of eight classes were used, including the background class. Four classes were assigned to semantic segmentation: maxillary sinus, maxilla, mandibular canal, and mandible. Three classes were assigned to instance segmentation: normal tooth, treated tooth, and dental implant.

**Figure 2 jcm-10-02577-f002:**
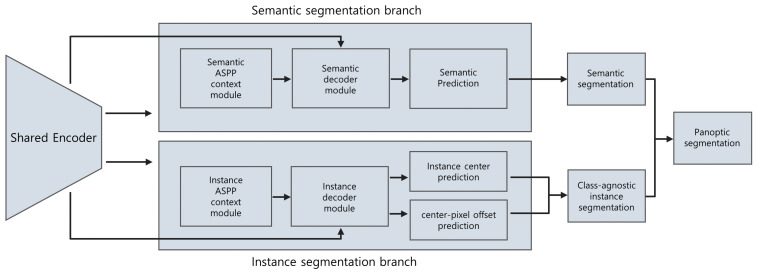
Overview of the model used in this study. Both semantic and instance segmentation branches use the same feature map and share the encoder. Each branch has its multiscale context module and decoder, both of which use ASPP. The instance segmentation branch predicts the center of mass for each instance and the offset vector between the center and each pixel. With the predicted center and the offset, the instance ID of each pixel can be determined, thereby resulting in class-agnostic instance segmentation. The final panoptic segmentation is obtained by fusing the results of the semantic segmentation and class-agnostic instance segmentation. ASPP, atrous spatial pyramid pooling.

**Figure 3 jcm-10-02577-f003:**
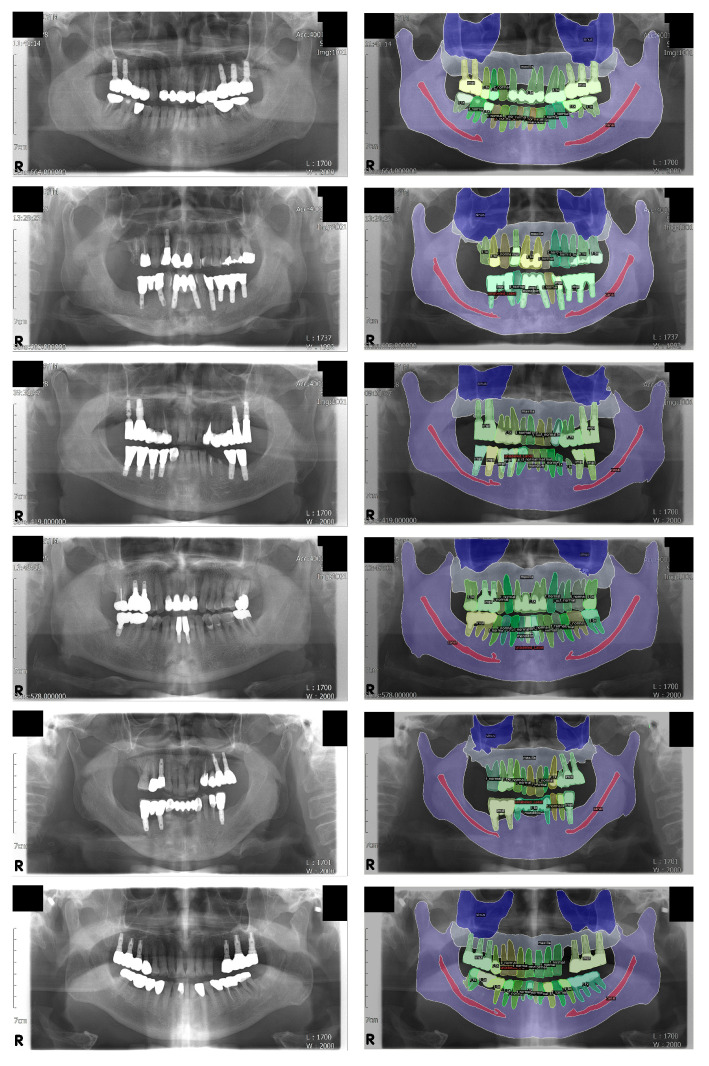
Visualization of the inference results. Some structures, including the mandibular canal and the medial wall of the maxillary sinus, are difficult to distinguish on the original panoramic radiographs but are fairly well segmented. Note that the normal tooth, treated tooth, and dental implant classes are individually segmented for each object; this was not possible with conventional semantic segmentation. Left: original input panoramic radiographs. Right: visualized result of the model’s inference.

**Figure 4 jcm-10-02577-f004:**
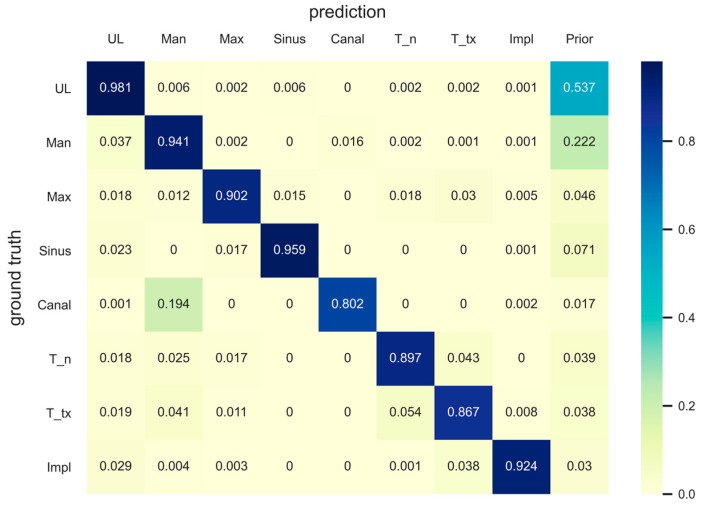
Confusion matrix. Each row and column represents the ground truth class and the predicted class, respectively. Each cell represents the ratio of the number of predicted pixels (column class) among the ground truth pixels (row class) to the number of ground truth pixels. Priors were computed for each row to represent the ratio of the number of pixels in the corresponding ground truth class to the total number of pixels. UL, unlabeled class; Man, mandible; Max, maxilla; T_n, normal tooth; T_tx, treated tooth; Impl, dental implant.

**Table 1 jcm-10-02577-t001:** Panoptic quality (PQ), segmentation quality (SQ), and recognition quality (RQ).

	PQ	SQ	RQ	N
Unlabeled	95.77	95.77	100.00	-
Mandible	89.73	89.73	100.00	-
Maxilla	82.77	82.77	100.00	-
Sinus	90.74	90.74	100.00	-
Canal	65.97	65.97	100.00	-
Stuff	85.00	85.00	100.00	5
Normal tooth	84.28	87.29	96.55	-
Treated tooth	57.13	85.69	66.67	-
Dental implant	77.34	87.89	88.00	-
Things	72.92	86.96	83.74	3
All	80.47	85.73	93.90	8

N: the number of classes.

**Table 2 jcm-10-02577-t002:** Intersection over union (IoU) and instance-level IoU (iIoU) for each class.

	IoU	iIoU
Unlabeled	0.954	-
Mandible	0.898	-
Maxilla	0.812	-
Sinus	0.898	-
Canal	0.639	-
Normal tooth	0.727	0.744
Treated tooth	0.656	0.611
Dental implant	0.775	0.827
Average	0.795	0.727

**Table 3 jcm-10-02577-t003:** Intersection over union (IoU) and instance-level IoU (iIoU) for each category.

	IoU	iIoU
Unlabeled	0.954	-
Bone	0.886	-
Sinus	0.898	-
Canal	0.639	-
Tooth	0.895	0.890
Average	0.854	0.890

Bone: a category including maxilla and mandible. Tooth: a category including normal tooth, treated tooth, and dental implant.

**Table 4 jcm-10-02577-t004:** Average precision (AP) for all the “thing” classes.

	AP (50)	AP (all)
Normal tooth	0.772	0.520
Treated tooth	0.490	0.316
Dental implant	0.714	0.414
Average	0.658	0.417

AP (50): AP at IoU threshold 0.5. AP (all): AP averaged across all intersection over union (IoU) thresholds.

## Data Availability

Restrictions apply to the availability of these data. Data used in this study were obtained from Seoul National University Dental Hospital and are available with the permission of the Institutional Review Board of Seoul National University Dental Hospital, Dental Life Science Research Institute.

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
