# Peer review of "Panoptic Segmentation on Panoramic Radiographs: Deep Learning-Based Segmentation of Various Structures Including Maxillary Sinus and Mandibular Canal"

_jcm, 2021, doi:10.3390/jcm10122577_

Round 1

Reviewer 1 Report

Very interesting application of panoptic segmentation of dental orthopanograms. The manuscript is well written but difficult to follow at time. The methodology is sound with good statistical analysis. However, the results were less impressive and the CNN had difficulty segmenting structures, notably the inferior alveolar canal which is a structure primordial importance. We would recommend the authors to compare the performance of their neural network to clinicians to show the clinical relevance of the system.

Reviewer 2 Report

  1. The panoramic radiographs :- what machine /equipment was used for taking xrays?, -what exposure conditions were taken?, -is the same person made all of included into the study OPG?positioning conditions?
  2. Could you give more details about exclusion conditions? ex. if chronic sinusitis should be excluded from the study ? why? ...
  3. You missed information about patients age,sex,etc... it seems important for homogeneity of samples included into your study
  4. The 51 OPG -is it enough for neural network models to be trained?what criteria did you have dividing OPG into 3 groups( training,validation and test)?
  5. Have you consider point/cropp segmentation method (manual) for anatomical points important for segmented structures ( mandibular canal -for ex. mental foramen,buccal foramen)? in your opinion might it improve neural network models training/learning?
  6. What are the weak points of the panoptic segmentation method from your study?

Reviewer 3 Report

The article entitled "Panoptic Segmentation on Dental Panoramic Radiographs: Deep Learning-based Segmentation of Various Structures including Maxillary Sinus and Mandibular Canal"  illustrated an automated method that detects the various structures present in the panoramic radiographs. 

The article was well written, the introduction and methods were well described, however, the major flaw is the argument, therefore, the article is rejected. 

For me, it's not imaginable that a clinician can't read correctly the panoramic radiographs. The individuation of anatomic structure is the basis to obtain a correct diagnosis and therapeutic treatments. Therefore, the student that did not read the panoramic radiographs can not obtain the degree.  Moreover, the panoramic radiograph represents the first step of the diagnostic tool. It is used only to have a panoramic view of the oral cavity and to individuate possible cyst, the position of the third molar, and other anatomic anomalies. Actually, the CBCT is the radiographic tool used to plan oral and maxillofacial surgery thanks to the 3-dimensional view of the anatomic structures. Therefore, the use of a panoramic radiograph presented limitations in the clinical diagnostic.

Author Response

Comments on the suggestions by the reviewers

We greatly appreciate your positive review and advice on our paper. We provide response to your comment below. In accordance with the guidelines presented, the responses we have written are in red.

Point 1:

The article entitled "Panoptic Segmentation on Dental Panoramic Radiographs: Deep Learning-based Segmentation of Various Structures including Maxillary Sinus and Mandibular Canal"  illustrated an automated method that detects the various structures present in the panoramic radiographs.

The article was well written, the introduction and methods were well described, however, the major flaw is the argument, therefore, the article is rejected.

For me, it's not imaginable that a clinician can't read correctly the panoramic radiographs. The individuation of anatomic structure is the basis to obtain a correct diagnosis and therapeutic treatments. Therefore, the student that did not read the panoramic radiographs can not obtain the degree.  Moreover, the panoramic radiograph represents the first step of the diagnostic tool. It is used only to have a panoramic view of the oral cavity and to individuate possible cyst, the position of the third molar, and other anatomic anomalies. Actually, the CBCT is the radiographic tool used to plan oral and maxillofacial surgery thanks to the 3-dimensional view of the anatomic structures. Therefore, the use of a panoramic radiograph presented limitations in the clinical diagnostic.

Response 1:

Thank you for your positive response and feedback. We agree with you that there are many cases where only using panoramic radiographs does not guarantee the correct diagnosis and treatment planning. On top of panoramic radiographs, periapical radiographs or CBCT should be taken if it is necessary. As you already mentioned, the panoramic radiograph is the first step of diagnosis in many cases. Moreover, dental clinicians should deal with many patients, and sometimes it is not easy to maintain the focus in reading many radiographs. Thus, we believe an automated segmentation system of panoramic radiographs can assist dental clinicians even if they need additional information via periapical or CBCT.

However, we agree with you again that the use of only panoramic radiographs has limitations. We hope this study can become a background of future studies for developing and evaluating a more generalized neural network or an integrated system that can use various kinds of imaging and clinical information altogether.

Round 2

Reviewer 2 Report

Thank you for your comments. In my opinion, the study should be continued to obtain more data. After the revision you made, it has been suggested to reconsider for publication.

Reviewer 3 Report

The authors have responded to the main question. However, I have the same doubts regarding the usefullness of the method. The article does not present serious flaws, therefore, could be published in this form.